# Early Spirometry Following Bronchoscopic Lung Volume Reduction with Endobronchial Valves

**DOI:** 10.3390/jcm11020440

**Published:** 2022-01-15

**Authors:** Pascal Bezel, Jasmin Wani, Gilles Wiederkehr, Christa Bodmer, Carolin Steinack, Daniel P. Franzen

**Affiliations:** 1Department of Pulmonology, University Hospital Zurich, Raemistrasse 100, 8091 Zurich, Switzerland; p.bezel@bluewin.ch (P.B.); gilles.wiederkehr@hirslanden.ch (G.W.); christa.bodmer@usz.ch (C.B.); carolin.steinack@usz.ch (C.S.); daniel.franzen@usz.ch (D.P.F.); 2Department of Pulmonology, Hirslanden Clinic St. Anna, St. Anna-Strasse 32, 6006 Luzern, Switzerland

**Keywords:** spirometry, pneumothorax, bronchoscopic lung volume reduction, endobronchial valves, emphysema

## Abstract

Bronchoscopic lung volume reduction (BLVR) by endobronchial valve (EBV) implantation has been shown to improve dyspnea, pulmonary function, exercise capacity, and quality of life in highly selected patients with severe emphysema and hyperinflation. The most frequent adverse event is a pneumothorax (PTX), occurring in approximately one-fifth of the cases due to intrathoracic volume shifts. The majority of these incidents are observed within 48 h post-procedure. However, the delayed occurrence of PTX after hospital discharge is a matter of concern. There is currently no approved concept for its prevention. Particularly, it is unknown whether and when respiratory manoeuvers such as spirometry post EBV treatment are feasible and safe. As per standard operating procedure at the University Hospital Zurich, early spirometry is scheduled after BLVR and prior to the discharge of the patient in order to monitor treatment success. The aim of our retrospective study was to investigate the feasibility and safety of early spirometry. In addition, we hypothesized that early spirometry could be useful to identify patients at risk for late PTX, which may occur after hospital discharge. All patients who underwent BLVR using EBVs between January 2018 and January 2020 at our hospital were enrolled in this study. After excluding 16 patients diagnosed post-procedure with PTX and four patients for other reasons, early spirometry was performed in 61 cases. There was neither a clinically relevant PTX during or after early spirometry nor a late PTX following hospital discharge. In conclusion, we found early spirometry, conducted not sooner than three days following EBV treatment, to be feasible and safe. Furthermore, early spirometry seems to be a useful predictor for successful BLVR, and it may help to decide whether a patient can be discharged. Given the small sample size and the retrospective design of our study, a prospective study that includes routine chest imaging after early spirometry to definitively exclude PTX is needed to recommend early spirometry as part of the standard protocol following EBV treatment.

## 1. Introduction

Chronic obstructive pulmonary disease (COPD) is a chronic inflammatory disorder of the respiratory tract with persistent airflow limitation, mucociliary dysfunction, and, eventually, irreversible and progressive destruction of the lung parenchyma leading to emphysema. Patients with advanced COPD and severe emphysema often remain symptomatic due to hyperinflation despite optimal medical treatment. In emphysema patients, hyperinflation, due to loss of pulmonary elastic recoil, is the most relevant cause of dyspnea since respiratory mechanics, particularly as related to diaphragmatic function, are severely impaired. Illustratively, diaphragmatic flattening is easily visible in the lateral view of plain chest radiography. Correspondingly, spirometry reveals severe obstruction, ideally with the typical check valve pattern suggesting severe hyperinflation.

However, radiology and spirometry are only static representations of hyperinflation, which can be proved physiologically on the basis of increased residual volume (RV) using body plethysmography. Functionally, physical performance, as measured by the 6-min walking distance (6-MWD), and quality of life are inversely correlated with increased RV [1].

In highly selected patients, after all of the conservative treatment options, as recommended by the Global Initiative of Obstructive Lung Disease (GOLD), were implemented, lung volume reduction, either by surgery (LVRS) or by bronchoscopic techniques (BLVR), was shown to improve symptoms, pulmonary function, exercise capacity and quality of life in multiple randomized controlled trials [1,2,3,4,5,6,7]. Following BLVR introduction of one-way endobronchial valves (EBVs), deployed into the most diseased lung lobe, air can exit during expiration but not reenter in inspiration. This procedure eventually leads to atelectasis of the treated lobe, provided there is no collateral ventilation (CV) between the treated and untreated ipsilateral lobes (Figure 1). This atelectasis forms the basis of lung volume reduction with the above-mentioned consequences. The absence of CV is an unconditional prerequisite for the development of atelectasis and, thus, for successful treatment. Therefore, CV must be excluded radiologically and/or bronchoscopically by using the Chartis^®^ console (PulmonX Inc., Redwood City, CA, USA) prior to EBV treatment.

EBV treatment generally has a favorable safety profile with pneumothorax (PTX) as the most common complication, with a reported occurrence of between 7% and 29% [4,6,7,8,9]. PTX management options include observation only, chest tube insertion, EBV removal, or surgical intervention [10]. In general, PTX, if properly managed following EBV treatment, is not associated with long-term morbidity. However, because PTX with continuing air leak can progress to a life-threatening tension PTX, their prevention or at least early detection is of great importance [11,12]. Approximately 75% of EBV-associated PTX occurs within three days and 85% within five days following implantation [7]. Therefore, inpatient observation following EBV implantation is strongly recommended with a hospital stay of at least two nights [10,11]. However, there is a 15% residual risk of late PTX occurring five days or later after EBV treatment when patients are already at home or in rehabilitation [7]. Out-of-hospital PTX is potentially associated with a serious adverse outcome. Gompelmann et al. proposed a risk stratification to assess the PTX risk after EBV treatment [11]. However, the individual risk of a PTX remains difficult to predict, and there is no generally approved concept to predict the development of late PTX after hospital discharge reliably. Furthermore, the question of whether patients should have bed rest after EBV treatment to prevent PTX has not been settled.

Spirometry is a non-invasive measurement of dynamic lung volumes during forced expiration after full inspiration. During the spirometry maneuver, the lung is exposed to high shearing forces, which may provoke the occurrence of a PTX. We hypothesized that early spirometry after EBV treatment is safe and may identify patients at risk for late PTX. The aim of this retrospective study was to investigate the feasibility and safety of early spirometry following BLVR with EBVs.

## 2. Materials and Methods

### 2.1. Patients

Between January 2018 and January 2020, all emphysema patients who underwent BLVR using EBVs at the University Hospital Zurich were enrolled for this retrospective study. Re-interventions for EBV replacement after an unsuccessful first attempt were included as a separate intervention, as each intervention causes a separate risk for PTX. Exclusion criteria were age below 18 years, pregnancy, emergency setting, lacking ability to provide informed consent, post-interventional PTX prior to early spirometry, and previous thoracic surgery, including LVRS. All patient-related data, including demographic and clinical data as well as reports of bronchoscopy and spirometry, were drawn from patient record files. Approval from the Competent Ethics Committee was waived because of formal non-objection (BASEC-ID 2018-02038).

### 2.2. Patient Selection

All patients who were possible candidates for LVRS or BLVR were previously discussed in a multidisciplinary emphysema board with the participation of pulmonologists and thoracic surgeons, eventually considering the patient’s preference. The basis for decision-making included the most recent pulmonary function test (PFT) values from spirometry, body plethysmography, CO-diffusion measurement (DLCO), 6-MWD, morphological aspects in single-photon emission computed tomography (SPECT), and high-resolution chest computed tomography (HR-CT). Target lobe selection and analysis of interlobar fissure completeness as morphological proxies for the absence of collateral ventilation (CV) were provided using StratX^®^ (Version 3.2.0.0, PulmonX Inc., Redwood City, CA, USA). A reported fissure completeness score > 95% as determined by independent estimation of the operator was used to indicate the absence of CV. Bronchoscopic exclusion of CV was conducted using the Chartis^®^ console (PulmonX Corporation, Redwood City, CA, USA) and was performed in a separate intervention before possible BLVR. This procedure was followed routinely in patients with a fissure completeness score below 95% in the StartX^®^ analysis and optionally in patients with a fissure completeness score above 95% in the StartX^®^ analysis. CV positivity and thus exclusion from EBV treatment was assumed when the fissure completeness score was below 80% or when CV was evident by Chartis^®^. In patients with a fissure completeness score between 80% and 95%, but inconclusive Chartis^®^ measurement due to early collapsing airways or mucus obstruction of the catheter, the decision for or against EBV treatment was made on a case-to-case basis. Generally, EBV treatment was recommended when the radiological appearance of interlobar fissures seemed complete.

### 2.3. Bronchoscopy and Post-Interventional Care

Chartis^®^ measurement and EBV treatment were provided using flexible Olympus (Olympus, Tokyo, Japan) bronchoscopes (190 series) through a laryngeal mask or tracheal tube under general anesthesia. After exclusion of CV, the most diseased lobe according to StratX^®^ and SPECT was targeted for BLVR using Zephyr^®^ endobronchial valves (PulmonX Inc., Redwood City, CA, USA), based on a recent expert recommendation paper [13].

Immediately after termination of EBV implantation, in-table PTX was excluded by bedside chest sonography. If there were no signs of PTX, patients were extubated and transferred to the intermediate care unit, where PTX was further excluded by conventional chest radiography (X-ray) in a supine position. X-ray was repeated in case of suspected PTX or, routinely, on the following day in an upright position.

As per institutional standard operating procedure from 1 November 2017, post-interventional spirometry was scheduled towards the end of the hospital stay, ranging from earliest on the second post-procedure day (48 h after EBV implantation) until the eighth day following BLVR with EBVs (median 3 days post-BLVR), providing there was no previous occurrence of an early PTX after BLVR. The latter was an exclusion criterion since spirometry is contraindicated in patients with PTX. Routine chest imaging after early spirometry was not part of the hospital protocol.

The timing of spirometry allowed for clinical observation during at least one additional night with routine measurements of blood pressure, heart rate, and oxygen saturation. In addition, the patients were asked to report any symptoms suggesting possible PTX immediately. An X-ray after spirometry was performed only in case of clinically suspected PTX. A chest drainage set was always available for immediate treatment in case of a clinically relevant PTX.

### 2.4. Spirometry

Spirometry was performed using Spirostik (Geratherm^®^ Respiratory, Bad Kissingen, Germany, Software Blue Cherry^®^, Lenswood, Australia) according to the official ATS/ERS consensus with a maximum of six attempts [14]. However, a bronchodilation test was discouraged for the purpose of this study.

### 2.5. Outcome Measures

Baseline measurements (spirometry, body plethysmography, DLCO, 6-MWD, and radiology) were obtained within three months prior to BLVR. Early spirometry was scheduled as mentioned above. The first routine follow-up measurement (spirometry) was conducted one to four months after EBV treatment. The main outcome measure was PTX rate during and after early spirometry.

### 2.6. Statistical Analysis

All statistical analyses were performed using SPSS Statistics for Windows 25.0 (IBM, Armonk, NY, USA). Data were reported as the median and interquartile range (IQR) or as percentages, as appropriate.

Changes in spirometry values were analyzed using the Wilcoxon test for two dependent variables.

## 3. Results

### Pneumothorax

A total of 81 cases in 66 patients were enrolled in this study. Baseline characteristics of the 66 patients are shown in Table 1.

Post-interventional PTX occurring prior to the early spirometry was observed in 16 out of 81 cases (19.8%) after a median of one day following BLVR (IQR 1–3 days). These cases were excluded from the study. In addition, four cases were excluded for other reasons (Figure 2). After exclusions, 61 cases were included in the study. Early spirometry was performed in the 61 cases mentioned above after a median of three days (IQR 2–3 days) following EBV treatment. There was no occurrence of PTX following early spirometry, either during the in-hospital stay or in the outpatient setting afterward (Table 2).

Measurement of forced expiratory volume in the first second (FEV1) in early spirometry as compared to baseline and routine follow-up spirometry are presented in Figure 3. FEV1 already improved significantly at early spirometry (*p* < 0.001) compared to baseline.

## 4. Discussion

Post-procedural care after BLVR using EBVs varies among intervention centers. Due to the risk of PTX, in-hospital observation for at least 48 h post-procedure is generally recommended [10]. The occurrence of a late PTX after hospital discharge is still a matter of concern. There is currently no approved concept for its prevention. Particularly, it is unknown whether and when respiratory manoeuvers such as spirometry post EBV treatment are feasible and safe. In 2017 early spirometry was introduced in our hospital as part of the standard BLVR protocol in order to assess lung volumes post-procedure as a measure of early treatment success. Routine chest imaging thereafter was not part of the protocol.

In this study, we retrospectively evaluated the feasibility and safety of early spirometry after EBV treatment and also addressed the question of its possible use for the identification of patients at risk of developing late PTX after hospital discharge.

Our findings showed no clinically apparent PTX during or after early spirometry. Based on our data, early spirometry after BLVR performed on the day prior to discharge, but no sooner than two days post-procedure, seems feasible and safe. Furthermore, we observed that early spirometry might be a useful predictor for successful BLVR since FEV1 already improved significantly at early spirometry (*p* < 0.001) compared to baseline.

We have hypothesized that early spirometry may precipitate a pneumothorax owing to rupture of a weakened pulmonary element under tension or possibly trigger the expansion of a small pneumothorax that may not have yet become apparent on earlier imaging. By triggering a more obvious PTX, early spirometry may be potentially useful in the identification of those patients who are likely to develop a clinically relevant PTX post-discharge, thereby allowing for appropriate intervention while the patient is still under care in hospital. Since chest imaging was not part of the hospital protocol after early spirometry, we did not have data to exclude small PTXs post spirometry. In our opinion, a lung ultrasound or chest x-ray should necessarily be performed thereafter. In addition, a chest drainage set should always be available for immediate treatment of a clinically relevant PTX.

Without data on chest imaging after early spirometry and since none of the patients in our study developed a clinically relevant PTX after discharge, we are not able to draw any conclusion about whether or not early spirometry may be helpful in reducing the incidence of PTX following discharge. If this question could be answered in future studies, early spirometry may help to decide whether a patient can be discharged.

There were 16 PTX events occurring within three days following EBV treatment, precluding the use of early spirometry. The prevalence and onset of post-BLVR PTX in our study were comparable to previous publications [10,11].

Herzog et al. reported a lower rate of PTX in patients who had strict bed rest for 48 h after the procedure in combination with pharmacological cough suppression compared to patients with standard care [12]. These results are challenged by our study since spirometry is a relatively stressful test. However, we did not observe any clinically relevant PTX thereafter. In addition, the patients in our study had bed rest only for up to 12 h and were not treated with any cough suppressant drugs.

Previous studies showed that a primary (unrelated to a previous medical procedure) PTX is generally believed to be related to physical strain, coughing, or Valsalva maneuvers [15,16,17]. In contrast, Bense et al. found that only 2% of PTXs in their retrospective study were associated with previous exertion [18].

The main limitations of our study are the retrospective study design, the relatively small sample size, and the absence of any PTX event during or following early spirometry. Furthermore, a small (subclinical) PTX following early spirometry could not be excluded since there was no routine chest radiography or sonography after spirometry.

## 5. Conclusions

We report here for the first time that early spirometry following EBV treatment on the day before hospital discharge seems feasible and safe. Furthermore, we found that it is a useful predictor for a successful BLVR and may help to decide whether a patient can be discharged. Since spirometry potentially induces shearing stress to the lung, the risk for PTX remains, and therefore lung ultrasound or chest x-ray is highly recommended afterward. In addition, a chest drainage set should always be available for immediate treatment of a clinically relevant PTX. Given the retrospective study design and the small sample size, the role of early spirometry in anticipation of late PTX could not be defined conclusively. A further prospective study including routine lung imaging after early spirometry is needed to address the inclusion of early spirometry as part of the routine procedure after EBV treatment.

## Figures and Tables

**Figure 1 jcm-11-00440-f001:**
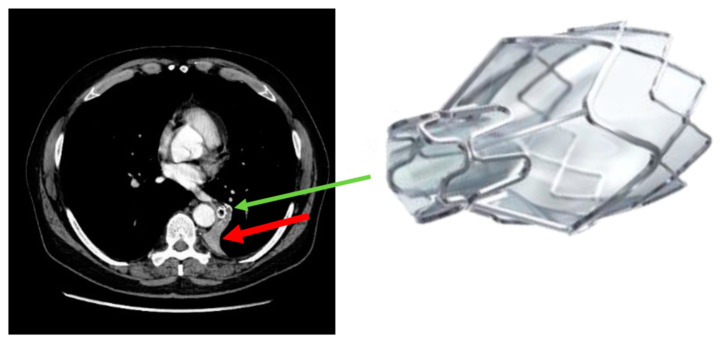
Atelectasis of the left lower lobe after endobronchial valve (EBV) implantation. The red arrow points at the atelectasis, the green arrow at the EBV.

**Figure 2 jcm-11-00440-f002:**
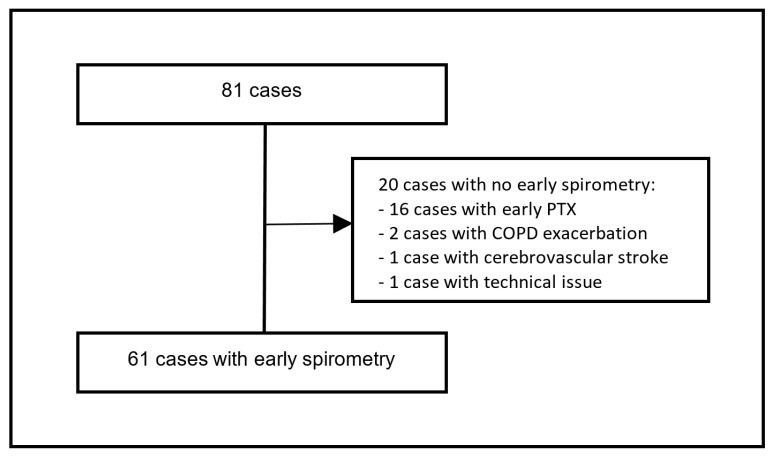
Cases flow chart (*n* = 81 cases total, *n* = 61 with early spirometry).

**Figure 3 jcm-11-00440-f003:**
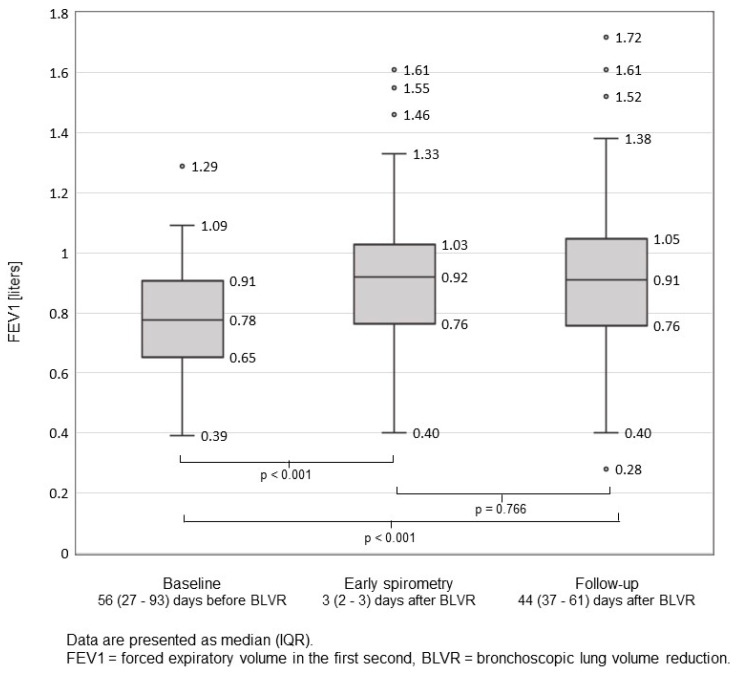
FEV1 at baseline, early spirometry, and follow-up after EBV treatment. Of the 61 cases with early spirometry, three were lost to follow-up spirometry.

**Table 1 jcm-11-00440-t001:** Baseline characteristics of all patients (*n* = 66).

**Demographic Data**		
Age, years	69.0	(62.9–74.4)
Male gender	30	(45)
Height	166	(162–171.8)
Weight	59	(52–71.8)
BMI	22.4	(19.3–24.6)
**EBV Implantation**		
Number of EBVs implanted per patient	5	(4–6)
Volume of target lobe *, mL	1510	(1277.5–1817)
Target lobe		
Left lower lobe	29	(43.9)
Left upper lobe	29	(43.9)
Right lower lobe	2	(3.0)
Middle lobe	1	(1.5)
Right upper lobe	6	(9.1)
**Baseline PFT and 6-MWD**		
FEV1, L	0.75	(0.56–0.91)
FEV1, % predicted	29	(23–34)
RV, L	5.09	(4.36–5.66)
RV, % predicted	230	(199–258)
RV/TLC, %	69	(63–74)
DLCO, mmol/kPa/min	2.4	(1.9–2.9)
DLCO, %	30	(25–38)
6-MWD, m	300	(225–368)

* assessed by StratX^®^. Data are presented as *n* (%) or median (IQR). Abbreviations: FEV1 = forced expiratory volume in the first second, RV = residual volume, TLC = total lung capacity, DLCO = diffusion capacity for carbon monoxide, mmol/kPa/min = millimolar/kilopascal/minute, 6-MWD = 6-min walk distance, PFT = pulmonary function test. EBV = endobronchial valves, mL = milliliters. Baseline PFT and 6-MWD were conducted at median 56 (27–93) days prior to BLVR.

**Table 2 jcm-11-00440-t002:** Post-interventional pneumothorax (*n* = 16).

**Post Procedural PTX before Early Spirometry**		
Same day	7	(43.8)
1st day	6	(37.5)
2nd day	2	(12.5)
3rd day	1	(6.3)
**Clinically Relevant PTX during or after Early Spirometry**		
PTX during Spirometry	0	(0)
PTX after Spirometry	0	(0)

Data are presented as *n* (%). PTX = pneumothorax.

## Data Availability

The data presented in this study are openly available upon reasonable request.

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
