# Peer review of "Early Spirometry Following Bronchoscopic Lung Volume Reduction with Endobronchial Valves"

_jcm, 2022, doi:10.3390/jcm11020440_

Round 1

Reviewer 1 Report

The study is novel and somehow interesting, yet I a have major concerns:

-firstly it is not clear to me the role of spirometry in the study as it is a diagnostic test and not a challenge; if you perform a challenge it should be performed in the same way and this is very different from a diagnostic test; moreover, and most important, the research hypothesis is not clearly stated and therefore conclusion not fully supported by data

-moreover pneumothorax is not primarily a functional defect but an anatomical alteration and imaging is key (why the authors did not used ultrasound after spirometry?)

-too many figures puzzle the readers and interrupt the reading flow; please focus on the main results.

-study limitation are not sufficiently acknowledged; also BLVR is a technique for highly selected patients and has many risks; this is not clearly stated (from the abstract to introduction); a

-references are not correct (e.g. ref 13 in the text is 14 in the references)

I think this study, considering also the small sample size, could be only used to generate hypothesis and not to modify clinical practice as mentioned by authors. I suggest to focus on spirometric results, which should be improved by an extensive evaluation (e.g. DLCO) as predictors and not as a challenge.

Reviewer 2 Report

Dear Editor and Authors,

Thank you for asking me to review this manuscript titled “Early Spirometric Challenge Test Following Bronchoscopic Lung Volume Reduction with Endobronchial Valves” by Drs Bezel and Wani and their colleagues from the University Hospital Zurich, in Zurich, Switzerland.

In this single center prospective single arm study the ability of a post-discharge spirometric challenge test to predict the incidence of late pneumothorax formation was evaluated in 61 patients undergoing bronchoscopic lung volume reduction (BLVR) with the use of endobronchial valves.

This is a well written and presented study using good and easily understood language. It is well structured and illustrated.

I do have a couple of comments/queries/suggestions for the authors:

  1. In my experience a number of thoracic surgeons utilizing BLVR advocate the use or peri-operative inserted prophylactic small bore (20 - 24 Fr) chest drains while the patient is under anesthesia. The drain is subsequently removed at patient discharge at 2 days. I would suggest this option may also be added as pneumothorax management (lines 80 - 82) and as support of their original premise i.e. “because we can not predict which patient  undergoing BLVR will develop a post-procedural pneumothorax, some thoracic surgeons advocate the use or peri-operative inserted prophylactic small bore (20 - 24 Fr) chest drain which is removed at discharge. However, this is not always feasible and presupposes familiarity with chest drains and their management”.
  2. I am worried that individual patient consent was waived by the institution’s ethics committee considering that the undergoing of a spirometric challenge has the potential to induce a pneumothorax in these patients and thus affect their health status! What is formal non objection and how is this ethically justified? In one part of the manuscript the authors mention that approval was waived and in the end that informed consent was obtained by all patients, how could patients give consent for a not approved study!!  This all needs clarification.
  3. Was Chartis assessment performed peri-operatively (this is my personal experience and preference) and then accordingly the patient underwent EBLV or was it done as a separate procedure? It is not clear in the text (patient selection – lines 127-128).
  4. If the purpose of the spirometry challenge was to exclude the formation  of a pneumothorax, why was not a routine chest x-ray not performed afterwards or the next day? Why did the authors rely only on clinical presentation? Could a bedside ultrasound not be performed at least? This looks suspiciously as a study extrapolation i.e. the authors performed the spirometry measurements for some other purpose (maybe clinical, maybe to see treatment efficacy or maybe as part of another study) and have extrapolated this new analysis using the previous/available data i.e. this is not the original study concept/hypothesis! I say this because the lack of a post spirometry CXR is a methodological limitation and needs to be explained!!
  5. My assumption above is further supported by the fact that no formal power analysis and sample size calculation was performed prior to study enrollment to confirm the minimal sample needed to provide statistically meaningful results (this is routine practice on all prospective protocols and part of methodology)!! Was the 61 patient enrolled (which by the way did not sign a prospective patient consent) adequate? This needs to be mentioned as a methodological limitation. This seems to this reviewer more and more as a retrospective data analysis of a non-widely established, departmental based clinical practice that it is now been attempted to be presented as a prospective study!! The gist of the aim of the authors with this manuscript is line 245: “Positive likelihood for early spirometry for predicting of a three months success”.
  6. I don’t understand how the StratX and Chartis measurements are relevant in the formation of a pneumothorax (table 2). This seems like superfluous data that add nothing to the main concept of the paper, are irrelevant and act only as page filler!! I suggest that they are eliminated.
  7. The authors present the number of EBV implanted and the location/lobes involved. However, was an multivariable analysis performed to correlate the number of EBVs, lobe(s) and volume to pneumothorax formation? Why was such an analysis not performed? Was the low number of patients in the sample and the incidence of pneumothorax a factor?  
  8. Why were there 20 cases with no early spirometry which were excluded from the analysis (figure 6) if this was a prospective study?  
  9. Why was there variability (median of 3 days) in the performance of the early spirometry if there was a pre-determined set time methodology (post-procedure day)?
  10. Why are the baseline and follow up (really at which follow up was spirometry performed? At 1, 2, 3 or 4 months – or at every appointment? And if so which appointment’s spirometry results were used to calculate the change/improvement in PFT? The best one? The latest one – when BLVR maybe failing) PFT data presented if this is a study about post-procedural pneumothorax? What value do they add to the analysis?
  11. I don’t understand table 4! No patient developed a pneumothorax after Early Spirometry? Why is timing of PTX after early spirometry all zeros? Therefore, the hypothesis of the study is wrong – early spirometry does not produce a pneumothorax. All 16 patients which developed pneumothorax would have developed it irrespectively of the spirometry. Yes?
  12. If these 16 patients developed pneumothorax prior to early spirometry, did they undergo spirometry at the end since it is contraindicated in this setting??
  13. Drainage time is reported as a median of 9 days with a range of 4.5 to 9.5, how is this possible? It means that most patients had a drain for at least 9 days!!
  14. Why did 3 patients undergo surgery for their pneumothorax?
  15. Of the 3 patients undergoing surgery for their post-intervention pneumothorax, what type of surgery was performed?

In conclusion this study is really convoluted and mixed up, I believe that the authors did not plan it to develop in the form presented. The presented hypothesis for example, is not supported by the results (the incidence of pneumothorax post early spirometry is zero!), I believe someone’s clinical practice includes the “not routinely done”/”non advocated” approach of early spirometry evaluation post BLVR to see “his/hers early efficacy” and at the end they decided to gather up their patients and results, analyze them and present them!! There is nothing wrong with that but it does not seem as a well designed prospective study!!

In addition, there has been significant weight placed on the early spirometry results (which as mentioned seems to be the actual intent) and not so much on the hypothesis about “predicting post procedural pneumothorax!”

Finally, the conclusion is not supported by the evidence. For example how can the authors claim that “Early spirometric challenge between two and eight days following EBV treatment is safe and can likely be used to avoid the occurrence of a late (out-of-hospital) and potentially life-threatening PTX” since they had none occur following spirometry.

Overall, I feel this work needs some major re-working.   

Reviewer 3 Report

This manuscript is well written regarding scientific aspects and format.

- The sample size is not high enough to reliably state that early spirometric challenge after bronchoscopic lung volume reduction with valves can predict late pneumothorax. However, this weakness of the study has been mentioned by authors.

Author Response

Thank you very much for your review and comment. With our changes in the manuscript we hope to have improved the presentation of our study to the satisfaction of all reviewers. Additionally we had the manuscripted prove read by a native english speaker (James Keel MD). We added him in our acknowledgement.